# Antennal and Behavioral Responses of *Drosophila suzukii* to Volatiles from a Non-Crop Host, *Osyris wightiana*

**DOI:** 10.3390/insects12020166

**Published:** 2021-02-15

**Authors:** Yan Liu, Zhihao Cui, Mi Shi, Marc Kenis, Wenxia Dong, Feng Zhang, Jinping Zhang, Chun Xiao, Li Chen

**Affiliations:** 1College of Life Science, Institute of Life Science and Green Development, Hebei University, Baoding 071002, China; lyliuyan87@163.com (Y.L.); zhihaocui2020@163.com (Z.C.); 2Plant Protection College, Yunnan Agricultural University, Kunming 650201, China; sm980314@126.com (M.S.); dongwenxia@163.com (W.D.); 3State Key Laboratory of Integrated Management of Pest Insects and Rodents, Institute of Zoology, Chinese Academy of Sciences, Beijing 100101, China; 4International Center for Applied Biosciences (CABI), CH-2800 Delémont, Switzerland; m.kenis@cabi.org; 5MoA-CABI Joint Laboratory for Bio-safety, Institute of Plant Protection, Chinese Academy of Agricultural Sciences, Beijing 100193, China; f.zhang@cabi.org (F.Z.); j.zhang@cabi.org (J.Z.)

**Keywords:** invasive species, spotted wing drosophila, GC-EAD, olfactory response, field trapping

## Abstract

**Simple Summary:**

The spotted-wing drosophila *Drosophila suzukii*, native to Southeast Asia, has become a new threat for cultivated and wild soft-skinned fruits in both their native and invaded areas. The attractiveness of volatiles from fresh ripening and ripe fruits of cultivated crops, such as raspberry, blueberry, strawberry, and cherry, has been extensively studied. To date, however, little attention has been paid to the non-crop host of *D. suzukii*. In this study, we collected headspace volatiles from ripe fruits of a wild non-crop host, *Osyris wightiana*, and identified seven compounds attractive to *D. suzukii* from the fruit volatiles. A blend of the seven active compounds captured a significant amount of female and male *D. suzukii* in field, suggesting that the blend can be used to develop an attractant for management and monitoring of *D. suzukii*.

**Abstract:**

*Drosophila suzukii* (Diptera: Drosophilidae) infests a variety of commercial fruits, including cherries and other soft-skinned fruits. After the cropping season of most cultivated crop hosts, it heavily infests the fruit of a wild host-plant, *Osyris wightiana* in southwest China. Here, we employ gas chromatography-electroantennographic detection (GC-EAD) together with behavioral bioassays and a trapping experiment to identify volatile semiochemicals emitted by *O. wightiana* that are involved in *D. suzukii* attraction. GC-EAD recordings of *D. suzukii* antenna showed responses to 13 compounds, including *α*-pinene, 3-methylbutyl acetate, 2-hexanol, *E*-*β*-ocimene, Z-3-hexenol, *β*-caryophyllene, *α*-humulene, and six unidentified compounds. The flies were attracted by seven individual EAD-active compounds at low doses (0.01 and 0.1 μg), but were repelled at high doses (10 and 100 μg). In a similar manner, a blend of seven EAD-active compounds at low doses (0.1 and 1 μg) was attractive to female flies, but became repulsive at high doses (10 μg). The low dose of the blend was as attractive as the fruit volatiles, although both were less attractive than the fruits. The blend attracted both female and male *D. suzukii* and other *Drosophila* flies. The percentage of *D. suzukii* out of all flies captured by the blend was significantly greater than that captured by the control. These results indicate that the EAD-active volatile compounds emitted by fruits of *O. wightiana* play an important role in *D. suzukii* attraction, and have the potential to be used for management of *D. suzukii*.

## 1. Introduction

The spotted wing drosophila, *Drosophila suzukii* Matsumura (Diptera: Drosophilidae), is native to Southeast Asia, and has been a worldwide invasive pest species to soft-skinned and stone fruits since 2008 [1,2]. This highly polyphagous pest infests many economically important fruit crops, such as apricots, berries, cherries, figs, grapes, kiwis, nectarines, peaches, pears, persimmons, plums, and pluots [3,4,5].

Wild growing crop species, such as blueberry and blackberry, are preferred hosts of *D. suzukii* [6,7,8]. Additionally, *D. suzukii* is known to attack a wide variety of wild/ornamental non-crop plants [9]. More than 150 non-crop plants belonging to 27 families have been determined to be suitable hosts of *D. suzukii*, providing a rich food resource throughout the year [6,10,11,12,13].

*Osyris wightiana* Wall (Santalaceae) is an evergreen shrub or short tree, 2 to 5 m in height in southwest China. It grows on rocky slopes and sparsely distributes in hilly land. Its fruit is similar in appearance, size, and texture to the common cherry (*Cerasus yunnanensis* (Franch.) Yu et Li, Rosaceae), but has a bitter taste. The fruiting season of *O. wightiana* is in late September to December and cultivation of crop fruits generally ends in September. Therefore, *D. suzukii* may migrate from cultivated crops to neighboring *O. wightiana* bushes and continue infestation before overwintering as adults in reproductive diapause. The heavy infestation of the fruits of *O. wightiana* between years increases the number of overwintering reproductive flies. Up to 80% of ripe fruits can be infested by *D. suzukii*, and usually 35 larvae are present in a fruit (0.6 g on average) (unpublished data). *Osyris wightiana* appears to act as an alternative host source for *D. suzukii* when crop hosts are not available, which may allow for an early-season rapid increase in the fly population in crop fields.

Like most insects, *D. suzukii* rely on olfactory detection of volatile compounds released from host plants to search for suitable feeding and oviposition sites [14,15]. For obvious reasons, much attention has been paid to identification of behaviorally attractive volatile compounds from crop hosts for *D. suzukii* [14,16,17,18], whereas no emphasis has been placed on non-crop hosts. To identify potential attractive compounds from non-crop host for development of a species-specific lure to manage this pest, we investigated the attraction of mated female *D. suzukii* to volatiles from ripe fruits of *O. wightiana*. Specifically, the main objectives of this study were to: (1) assess the behavioral response of mated female *D. suzukii* to volatiles from ripe fruits of *O. wightiana*; (2) identify volatile compounds that can be detected by mated female *D. suzukii*; and (3) evaluate the attractiveness of individual antennally active compounds and their mixture in a series of laboratory two-choice bioassays and field trapping experiments.

## 2. Materials and Methods

### 2.1. Insects

The colony of *D. suzukii* used in this study was obtained from infested fruits of bayberries, *Myrica rubra* (Lour.) Sieb. et Zucc. in Yunnan Province, China in May 2016. The flies were reared on peeled bananas in an incubator chamber at 24 ± 1 °C, 65 ± 5% R.H., and 14:10 h L:D.

Newly-enclosed flies were kept in a cage (30 × 30 × 30 cm) with access to food and water, and were allowed to mate for 3 days. Adult flies were starved for approximately 24 h prior to experiments. Only five-day-old mated females were used in behavioral and electrophysiological experiments.

### 2.2. Collection of Fruit Volatiles

Newly collected *O. wightiana* fruits at ripe stage (about 0.6 kg) from Changchong Mountain, Kunming, Yunnan Province were enclosed in a glass vessel (6 cm diam., 20 cm high). Air entering the vessel was drawn through a filter filled with activated charcoal. Air was pulled out a vacuum pump (QC-1S; Beijing Municipal Institute of Labor Protection, Beijing, China) from the vessel at 400 mL/min through an adsorbent tube (5 mm internal diam.; 500 mg Porapak-Q, 80/100 mesh, Supelco, Bellefonte, PA, USA) for 3 h. All connections were made with Teflon tubing. Volatiles were eluted from the Porapak-Q trap with 3 mL of hexane (98%, Merck, KGaA, Darmstadt, Germany). The volatile samples from 4 collections were combined and concentrated to 500 μL under a mild N_2_ stream for chemical analysis, electrophysiological experiments, and behavior experiments. A second set of 4 collections were conducted as described above for quantitative analysis by GC-FID. Nonyl acetate [19] was added to volatile samples of each collection as the internal standard for subsequent quantification.

### 2.3. Coupled Gas Chromatography-Electroantennographic Detection (GC-EAD)

The *O. wightiana* fruit volatile sample (2 μL) was injected in a Shimadzu GC-2010plus equipped with a DB-Wax column (30 m × 0.25 mm inner diameter × 0.25 μm film thickness; Agilent) and interfaced with the EAG apparatus. Helium (2.0 mL/min) was used as the carrier gas. The oven temperature was held at 40 °C for 2 min and programmed at 5 °C/min to 120 °C, then 15 °C/min to 240 °C and held for 4 min. The column effluent was split at a 1:1 ratio, with one part to a heated line into a humidified airstream (400 mL/min) which was directed at the antenna preparation, and the other to the Flame Ionization Detector (FID) of GC. Electroantennogram (EAG) recordings were made using Ag-AgCl glass microelectrodes filled with a Ephrussi-Beadle Ringer solution. The intact female fly was immobilized by cotton inside a pipette tip, while its head stretched out. The recording electrode was inserted into the distal region of the terminal antennal segment, while the reference electrode was positioned into the base of the antenna. The signals generated by the EAD and FID were passed through a Syntech IDAC-2 high-impedance amplifier and analyzed with Syntech GC-EAD software. FID peaks that elicited EAD responses for at least three runs were considered electrophysiologically-active and marked for identification by coupled gas chromatography-mass spectrometry (GC-MS).

### 2.4. Chemical Analyses

The volatile sample was analyzed with an Agilent 7890A GC coupled to a 5975C mass selective detector equipped with a polar DB-Wax column. Helium (1.0 mL/min) was used as carrier gas. The oven temperature was maintained at 40 °C for 4 min, and then increased to 60 °C at 1 °C/min, to 75 °C at 3 °C/min, to 180 °C at 5 °C/min, and subsequently to 240 °C at 10 °C/min, with a final 4-min holding time. The injector and transfer line temperatures were set at 230 °C and 250 °C, respectively. Mass spectra were obtained by using the electron impact (70 eV). The chemicals that were consistently EAD-active were tentatively identified by comparing their mass spectra with those of the NIST08 database. Their chemical identities were further confirmed by (1) comparing chromatographic retention times through co-injection with those of corresponding authentic standards, and (2) calculating Kovats Indices by comparing the retention times of the compounds of interest with those of the C7-C23 normal alkanes under the same GC-MS conditions.

GC-FID analyses were conducted using a Shimadzu GC-2010plus equipped with a DB-Wax capillary column in splitless mode. The oven temperature was programmed in accordance with GC-EAD recordings. Eighty μL of 0.1 μg/μL nonyl acetate was added to volatile samples of each collection (3 mL) as an internal standard for chemical quantification by comparing their peak areas.

### 2.5. Chemicals

All chemicals used in this work were purchased from Sigma-Aldrich (St. Louis, MO, USA), including nonyl acetate (99.0%), *α*-pinene (98%), 3-methylbutyl acetate (99%), 3-hexanol (97%), *β*-ocimene (90%), (*Z*)-3-hexenol (97%), *β*-caryophyllene (98%), and *α*-humulene (96%).

### 2.6. Y-Tube Olfactometer Bioassays

The behavioral responses of female *D. suzukii* to host fruits, volatile extract of host fruits and synthetic compounds were investigated in a Y-tube olfactometer (stem and arms: 15 cm and 10 cm long, respectively, 2 cm internal diam., and 60° Y angle). Two arms of the olfactometer were connected to two bulbs with an open end (φ = 2 cm). The open end of the arm slightly approaches into the bulb, which can partially prevent flies coming back to the main stem. The whole Y-tube olfactometer was housed inside a plastic box (43 × 30 × 18 cm) which was covered with an opaque black cloth to avoid visual distraction of the flies. Air entering the arms was pumped (atmospheric sampler) out the stem of the olfactometer at 200 mL/min. To test the response of mated *D. suzukii* females to host fruits, 25 intact fresh *O. wightiana* fruits were enclosed in one bulb as the other bulb was left empty (control). Ten females at a time were introduced at the entrance of the main stem. The number of flies that entered into the side arms was counted after 15 min. The flies that remained in the main stem after this time period were recorded as ‘no choice’. The olfactometer set-up was cleaned with ethyl alcohol and the glass parts were baked at 200 °C after every trial. There were 10 replicates for each treatment, and all flies were used only once.

The responses of mated *D. suzukii* females to fruit volatile sample, individual EAD-active compounds and their mixture at natural ratio (*α*-pinene:3-methylbutyl acetate:2-hexanol:*β*-ocimene:*Z*-3-hexenol:*β*-caryophyllene:*α*-humulene = 38:127:21:31:17:8:5) were conducted in the same manner. One glass bulb holding a filter paper disk (φ = 1.5 cm) loaded with 10 µL of the sample solution served as treatment while the other glass bulb holding a filter paper disk of the same size received 10 µL of hexane and served as control. All the synthetic compounds were prepared in hexane and diluted to 10, 1, 0.1, 0.01, and 0.001 μg/μL while the mixture was diluted with hexane to 1, 0.1, and 0.01 μg/μL.

### 2.7. Field-Trapping Experiments

Field-trapping experiments were conducted in an agrestal population of *O. wightiana* in Kunming, Yunnan Province (102.73° E, 25.04° N, altitude 1943 m) from 26 October to 7 November in 2020. The site was far away from commercial berry orchards, and there were agrestal weeds and shrubs mixed with the *O. wightiana* population. Most *O. wightiana* plants were about 2-meters tall. One hundred µL of the mixture of the 7 EAD-active compounds, at natural ratio mentioned above (0.01 μg/μL as Blend 1, and 0.1 μg/μL as Blend 2), or hexane (control) were loaded into an aboral bell-shaped rubber (length = 1.5 cm, Pherobio Technology Co., Ltd., Beijing, China). The lure was placed in a hole at the center of a blue sticky trap (25 cm, Tianxia Wuchong Pest Control Co., Ltd., Nanjing, China). All the traps were hung on *O. wightiana* tree branches 1~1.5 m above the ground, and 20 m apart.

A randomized complete block design was used for the field experiment. There were 9 replicate blocks, having two treatment traps and a control trap in each block. The numbers of female and male *D. suzukii* and other flies captured on each trap were counted at the end of the field trapping experiment (12 days in total).

### 2.8. Statistical Analyses

Data analysis was performed by SPSS 20.0 (SPSS Inc., Chicago, IL, USA). For the laboratory dual-choice bioassay, a significant difference between treatment and control was analyzed by Wilcoxon signed-rank tests (*p* < 0.05). The differences in trap catches and the calculated percentages of *D. suzukii* out of all flies captured were compared among blends and control treatments by the Tukey’s test at the 0.05 level.

## 3. Results

### 3.1. Bioactive Volatile Compounds Released by O. wightiana Fruits

Thirteen compounds in the volatile sample of *O. wightiana* fruits consistently elicited antennal responses in female *D. suzukii* (Figure 1). Seven of the EAD-active compounds were identified as *α*-pinene (1), 3-methylbutyl acetate (4), 2-hexanol (6), *E*-*β*-ocimene (9), *Z*-3-hexenol (10), *β*-caryophyllene (12), and *α*-humulene (13). These active components constitute 3.42% of the volatile sample (Table 1).

### 3.2. Dual-Choice Olfactometer Bioassays

Significantly more *D. suzukii* were attracted to fruits and the volatile sample of fruits of *O. wightiana* than the control (Figure 2).

The low doses of synthetic EAD-active compounds were found to be attractive, i.e., 0.01 and 0.1 µg for 2-hexenol, *E*-*β*-ocimene, *Z*-3-hexenol, and *α*-humulene, 0.1 and 1 µg for *α*-pinene, and 0.01 to 1 µg for 3-methylbutyl acetate and *β*-caryophyllene. When higher doses were tested, however, significant repellent activities were obtained for 1–100 µg of *Z*-3-hexenol and *α*-humulene, 10 and 100 µg of *α*-pinene and 3-methylbutyl acetate, and 100 µg of 2-hexanol, *E*-*β*-ocimene, and *β*-caryophyllene (Figure 2).

In the same manner, the blend of seven EAD-active compounds (0.1 and 1 µg) was significantly attractive to tested females at lower doses, but showed apparent repellency at a higher dose (10 µg) (Figure 2). Furthermore, the attractiveness of the seven-compound blend at 0.1 µg was similar to that of volatile sample of fruits.

### 3.3. Field-Trapping Experiment

The number of female and male *D. suzukii* captured by traps baited with the volatile blend was significantly higher than by control traps (Levene test: *df*_1_ = 2, *df*_2_ = 24, *p* = 0.246; *F* = 74.864, *p* < 0.0001). The percentages of *D. suzukii* out of all flies captured by the blend were significantly higher than that captured by hexane control traps (Levene test: *df*_1_ = 2, *df*_2_ = 24, *p* = 0.326; *F* = 111.654, *p* < 0.0001). Both blends and hexane control traps captured more males than females of *D. suzukii* (Figure 3). About half of the non-target flies captured were *D. melanogaster*.

## 4. Discussion

Several studies have reported on the olfactory responses of *D. suzukii* to volatile semiochemicals from intact fruits of cultivated host plants [14,16,17,18]. *Drosophila suzukii* appears to respond significantly to these volatiles. Here we reported the identification of active volatiles from *O. wightiana*, a preferred wild host plant of *D. suzukii*. A non-crop plant species was tested for the first time.

Herbivorous insects usually detect volatile compounds released by host plants at certain distance from the release source for host location and selection [19]. Our laboratory Y-tube tests demonstrated that mated female *D. suzukii* responded positively to *O. wightiana* fruits and their headspace volatile sample, indicating that olfaction plays a key role in *D. suzukii* host location [14,18,20]. Volatile compounds released by *O. wightiana* fruits that elicited antennal responses in female *D. suzukii* included terpenes (*α*-pinene, *E*-*β*-ocimene, *β*-caryophyllene, and *α*-humulene), an ester (3-methylbutyl acetate), and alcohols (2-hexanol and *Z*-3-hexenol). Alcohols, esters, and terpenes are among the most abundant compounds in Chinese bayberry [21,22], raspberry, blueberry, blackberry, strawberry, and cherry [18]. The six EAD-active compounds, *α*-pinene, 3-methylbutyl acetate, 2-hexanol, *E*-*β*-ocimene, *Z*-3-hexenol, *β*-caryophyllene, and *α*-humulene have been found from berry and cherry fruits [16,17,18,22], peach [23], grape [24,25], pear [25,26], apple [25], and apricot [27]. In the present study, *β*-caryophyllene elicited apparent GC-EAD response in *D. suzukii*. By contrast, no antennal response was recorded to *β*-caryophyllene, a predominant component released from Chinese bayberry in our previous study [16].

The percentage of responding flies and the attractive effect of the seven EAD-active compound blend at lower doses to *D. suzukii* were similar to those of the fruit volatile extract of *O. wightiana*, further indicating that the seven EAD-active compounds are key components from *O. wightiana* fruits responsible for attracting *D. suzukii*. The percentage of active components in the total amount of the sample analyzed by GC was 3.42% (Table 1), while over 95% of volatile components were not detected by the antennae of *D. suzukii* in our GC-EAD recordings. These results suggest that although fruits of *O. wightiana* emit a variety of volatile compounds that signal their identity and physiological state, only a small portion of these volatile compounds can be detected and used by *D. suzukii* for host location. These green leaf volatiles and species-specific odors may convey both general and specific information on fruits of *O. wightiana*. In this study, however, banana was used to rear *D. suzukii*. Laval experience on banana may affect olfactory preferences of adult *D. suzukii* to fruits of *O. wightiana*. Innate olfactory response in *D. suzukii* reared on artificial diet to volatiles of *O. wightiana* fruits awaits further investigation.

The seven individual EAD-active compounds and their blend displayed dose-dependent olfactory behavior in *D. suzukii*. The overall response pattern followed the same trend with attraction to low doses, indifference to intermediate dose and repulsion to high doses (Figure 2). This dose-dependent behavioral response pattern could be a common feature of olfactory perception in *Drosophila* [28]. For instance, six attractive odorants from ripen fruits, ethyl acetate, acetoin, butyl butyrate, ethyl hexanoate, ethyl 3-hydroxybutyrate, and phenylacetonitrile were attractive to *D. melanogaster* at lower concentrations, but aversive to them at higher concentrations [29]. Caged greenhouse experiments showed that traps baited with a lower dose (5.5 µL) of ethyl acetate captured more *D. suzukii* than that with higher doses (55 µL and 550 µL) [30]. In a recent report, we demonstrated that four EAD-active compounds released from Chinese bayberry, methyl (*E*)-3-hexenoate, methyl (*E*)-2-hexenoate, ethyl (*E*)-2-hexenoate and α-humulene were attractive to mated female *D. suzukii* at lower doses (0.01 and 0.1 μg), but showed repellency at higher doses (10 and 100 μg) [16]. The mechanism underlying the behavioral switch in response to higher concentrations of odorant in *D. melanogaster* has been well studied. The activation of two glomeruli, DM1 and VA2, was found to be responsible for the attraction of apple cider vinegar at low concentrations to *D. melanogaster*. A higher concentration of vinegar is less attractive to flies and excites an additional glomerulus, DM5 that independently mediates aversion. The activation of the aversive glomerulus DM5 may counterbalance the activation of the above two attractive glomeruli. The activation of DM1 or DM5 by any odor should be sufficient for concentration-dependent behavioral switch in *D. melanogaster* [28]. It needs to be tested whether *D. suzukii* has a similar brain structure or nervous system to *D. melanogaster*.

Field capture of both female and male *D. suzukii* confirmed the attraction of the seven EAD-active compound blend. Interestingly, the blend and control (hexane) captured significantly more males than females, same as sugar-vinegar-water solution for annual monitoring of *D. suzukii* (unpublished data). A reasonable interpretation for this difference was that the population of *D. suzukii* in Yunnan province was male-biased during the season of the field trapping experiment. Similarly, significantly more *D. suzukii* males were caught by the bayberry volatile blend than females in our previous study [16]. The seasonal sex bias in *D. suzukii* flies have been reported in previous studies from other regions. Rossi-Stacconi et al. [31] observed that male *D. suzukii* were more abundant than females in both spring and autumn in an Italian mountain region. Significantly higher numbers of male *D. suzukii* were captured in summer and autumn in the same area [2]. A strong male bias was observed in late autumn [7]. Furthermore, the percentage of *D. suzukii* captured by the blend was significantly higher than that captured by the control (hexane), indicating that the blend of volatiles derived from a wild host plant was selective in attracting *D. suzukii*. Moreover, the six unidentified EAD-active compounds could also contribute to attraction of *O. wightiana* fruits to *D. suzukii*. The identification of these EAD-active volatiles might allow the development of more selective and powerful attractant lures.

## 5. Conclusions

Field margins have been considered to play an important ecological role in providing alternative host resources, overwintering habitats, and refuge areas for insects [32,33]. High infestation rate (unpublished data) in *O. wightiana* shrubs suggested that *O. wightiana* can act as an alternative host source for *D. suzukii* during the post-harvest season of their cultivated host plants in Yunnan province because of its capability for long-distance dispersal across large geographic landscapes [34,35]. As a wild host plant, *O. wightiana* surrounding crop fields may play an important role in maintaining off-season *D. suzukii* populations [36]. Management tools such as host volatile compound-based attractants [37], food-based lures and traps [1], protein bait [38], or synthetic attractants [39] applied to field margins may reduce *D. suzukii* populations, especially if implemented in area-wide programs. Our study demonstrates that the fruit odors become less attractive and eventually repellent as their intensity is increased, suggesting that control of appropriate releasing rate of volatile mixtures is important for future development of lures used for management of *D. suzukii* in the field. Further investigations should test the effect of late season management of *D. suzukii* in *O. wightiana* habitats on its population size on the next-season crop cultivars.

## Figures and Tables

**Figure 1 insects-12-00166-f001:**
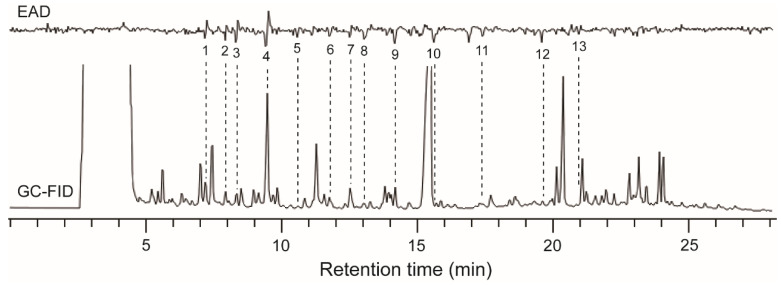
Gas chromatography-electroantennographic detection (GC-EAD) responses of female *Drosophila suzukii* to headspace volatile sample of *Osyris wightiana* fruits. The EAD-active compounds identified are listed in Table 1.

**Figure 2 insects-12-00166-f002:**
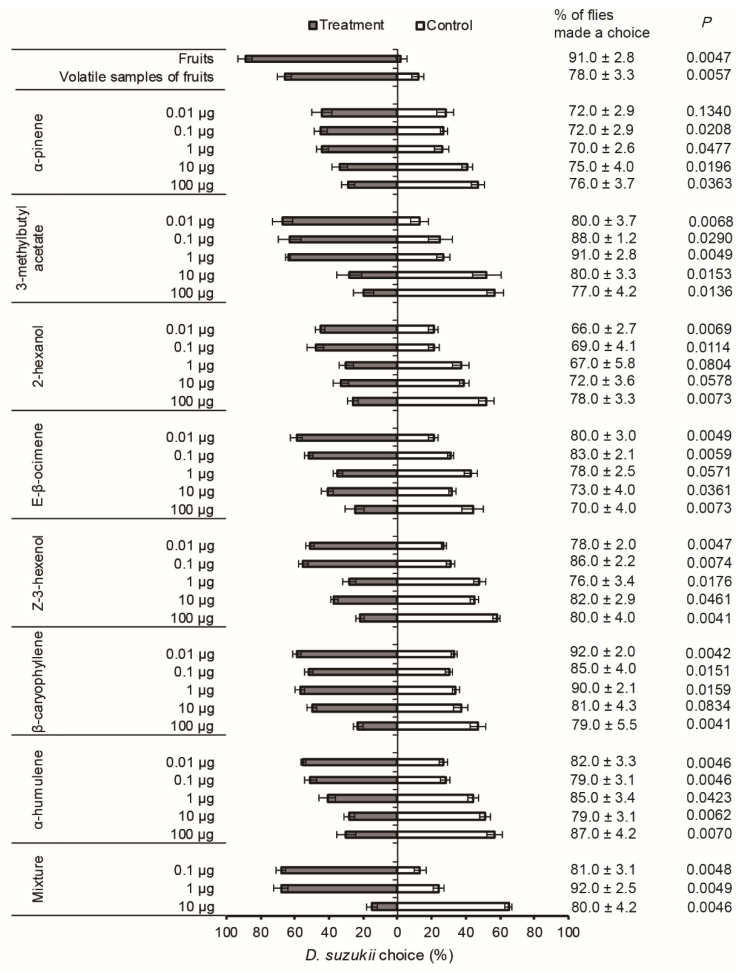
Olfactory responses of female *Drosophila suzukii* to *Osyris wightiana* fruits, the headspace volatile sample from fruits, individual EAD-active compounds and their mixture. N = 100 per treatment. The *p* value was calculated according to Wilcoxon signed-rank tests at the 0.05 level.

**Figure 3 insects-12-00166-f003:**
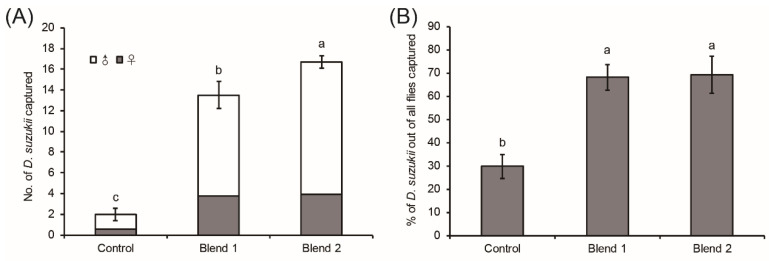
(**A**) Number (mean ± SE) of female and male *Drosophila suzukii* captured in traps (*N* = 9). (**B)** Percentage (mean ± SE) of *D. suzukii* captured out of all flies (*N* = 9). Control: hexane; Blend: a mixture of 7-EAD-active compounds of 0.01 µg/µL (Blend 1) and 0.1 µg/µL (Blend 2) at the natural ratio (100 µL). Different letters indicate significant differences among different treatments at the 0.05 level.

**Table 1 insects-12-00166-t001:** EAD-active compounds to female *Drosophila suzukii* emitted from ripe fruits of *Osyris wightiana*.

Peak No.	Compounds	RI ^a^		Relative Content (ng/µL)	Percentage ^b^ (%)
		Exp.	Lit.		
1	α-Pinene	1015	989–1052	2.7 ± 0.9	0.38 ± 0.13
2	Unidentified	1043	-	1.5 ± 0.3	0.21 ± 0.04
3	Unidentified	1063	-	1.3 ± 0.5	0.18 ± 0.07
4	3-methylbutyl acetate	1119	1102–1140	9.0 ± 1.6	1.27 ± 0.23
5	Unidentified	1167	-	0.4 ± 0.0	0.05 ± 0.00
6	2-Hexanol	1216	1192–1217	1.5 ± 0.5	0.21 ± 0.07
7	Unidentified	1246	-	0.7 ± 0.1	0.38 ± 0.05
8	Unidentified	1269	-	0.7 ± 0.1	0.10 ± 0.01
9	*E*-β-ocimene	1301	1232–1290	2.2 ± 0.7	0.31 ± 0.10
10	*Z*-3-hexenol	1369	1351–1390	1.2 ± 0.3	0.17 ± 0.04
11	Unidentified	1422	-	0.2 ± 0.1	0.03 ± 0.02
12	*β*-Caryophyllene	1556	1556–1615	0.6 ± 0.2	0.08 ± 0.03
13	*α*-Humulene	1676	1623–1705	0.4 ± 0.1	0.05 ± 0.01

^a^ RI means retention index on a DB-Wax column. Exp. Means RI calculated from the experiment. Lit. means RI found in the NIST Chemistry Web Book (http://webbook.nist.gov/chemistry/ (accessed on 13 February 2021)). ^b^ Percentage means the ratio of a peak out of all peaks from a volatile sample (N = 4).

## Data Availability

The data presented in this study are available on request from the corresponding authors.

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
