# Peer review of "Antennal and Behavioral Responses of Drosophila suzukii to Volatiles from a Non-Crop Host, Osyris wightiana"

_insects, 2021, doi:10.3390/insects12020166_

Round 1

Reviewer 1 Report

Dear editors and authors,

following you will find my comments to the revised manuscript “Antennal and Behavioral Responses of Drosophila suzukii to Volatiles from a Non-Crop Host, Osyris wightiana” submitted to Insects.

The second revision improved the manuscript a lot, but I have one important point for the Table 1:

  • Please give some more effort in the literature of RI range. You have given the full range of all columns shown at the NIST-web side. But you have to compare your measured RI with RI-lit. given for the column you used, here DB-wax or HP-Wax with column length used (by the way, this value is missing in M&M), than the lit.- range is much smaller. E.g. for alpha-pinene RI-lit. is than from 1011-1031 and for 2-hexanol it is 1211-1232, which is more accurate and supporting your results much stronger!! The range you are showing diminishes the excellent comparability of GC-measurements.

Thanks and best regards.

Author Response

Please give some more effort in the literature of RI range. You have given the full range of all columns shown at the NIST-website. But you have to compare your measured RI with RI-lit. given for the column you used, here DB-wax or HP-Wax with column length used, (by the way, this value is missing in M&M),than the lit.-range is much smaller. E.g. for α-pinene RI-lit. is than from 1011-1031 and for 2-hexanol it is 1211-1232, which is more accurate and supporting your results much stronger!! The range you are showing diminishes the excellent comparability of GC-measurements.

Response: The RI with same column (DB-wax column, 30 m × 0.25 mm inner diameter × 0.25 μm film thickness) was rechecked on NIST-website. The sentence “calculating Kovats Indices by comparing the retention times of the compounds of interest with those of the C(7-23) alkanes under the same GC-MS conditions” was added in M&M (2.4 Chemical analysis).

Reviewer 2 Report

Antennal and Behavioral Responses of Drosophila suzukii to Volatiles from a Non-Crop Host, Osyris wightiana by Yan Liu, Zhi-Hao Cui, Mi Shi, Marc Kenis, Wen-Xia Dong, Feng Zhang, Jin-Ping Zhang, Chun Xiao, and Li Chen

General comments

The study deals with the response of Drosophila suzukii flies to fruit odors and volatiles of the non-rop host, Osyris wightiana. Authors performed very well designed behavioral experiments in laboratory and in the field that went from intact fruit to isolated volatiles. They also included and electro-anntenograph analysis to determine the response to specific volatiles. Once identified, field experiment was carried out to determine the effects of some blends in the wild. The article is well written and experiments well designed and data well analysed. I like an agree with the conclusions they wrote. I just have some questions about the field experiment.

Particular comments

Line 156. Authors refer as orchard of Osyris wightiana. It is not an orchard since an orchard is an intentional plantation of trees or shrubs that is maintained for food production. Better refer to it as agrestal population. What was the tree height used to hang traps up?

Please mention the surface area where the field experiment was carried out. Was it a mixed species population. Was the experimental area close to commercial cherries orchards?

Line 170. “The differences in trap catches and the calculated percentages of D. suzukii out of all flies captured were…” Did the authors transformed the percent data? If did please mention it.

Author Response

General comments

The study deals with the response of Drosophila suzukii flies to fruit odors and volatiles of the non-crop host, Osyris wightiana. Authors performed very well designed behavioral experiments in laboratory and in the field that went from intact fruit to isolated volatiles. They also included and electro-anntenograph analysis to determine the responses to specific volatiles. Once identified, field experiment was carried out to determine the effects of some blends in the wild. The article is well written and experiments well designed and data well analyzed. I like an agree with the conclusions they wrote. I just have some questions about the field experiment.

Response 1: Thanks for positive comments.

Particular comments

Line 156. Authors refer as orchard of Osris wightiana. It is not an orchard since an orchard is an intentional plantation if trees or shrubs that is maintained for food production. Better refer to it as agrestal polulation. What was the tree height used to hang traps up?

Response 2: Yes, agrestal polulation was more suitable. The sentence was changed to “Field-trapping experiments were conducted in an agrestal population of O. wightiana in Kunming, Yunnan Province (102.73° E, 25.04° N, altitude 1943 m) from 26 October to 7 November in 2020”. The most trees are about 2-m high.

Please mention the surface area where the field experiment was carried out. Was it mixed species population. Was the experimental area close to commercial cherries orchards?

Response 3: The surroundings were added asThe site was far away from commercial berry orchards, and there were agrestal weeds and shrubs mixed with the O. wightiana population.

Line 170. “The difference in trap catches and the calculated percentages of D. suzukii out of all flies captured were…” Did the authors transformed the percent data? If did please mention it.

Response 4: The percentage of D. suzukii out of all flies captured were calculated, but not transformed.
